# Dysregulated Wnt Signalling in the Alzheimer’s Brain

**DOI:** 10.3390/brainsci10120902

**Published:** 2020-11-24

**Authors:** Nozie D. Aghaizu, Hanqing Jin, Paul J. Whiting

**Affiliations:** 1UK Dementia Research Institute at University College London, Cruciform Building, Gower Street, London WC1E 6BT, UK; hanqing.jin@ucl.ac.uk; 2ARUK Drug Discovery Institute (DDI), University College London, Cruciform Building, Gower Street, London WC1E 6BT, UK

**Keywords:** Wnt signalling, Alzheimer’s disease, neurodegeneration, APP processing, tau pathology, synapse degeneration, neuroinflammation, blood brain barrier

## Abstract

The Wnt signalling system is essential for both the developing and adult central nervous system. It regulates numerous cellular functions ranging from neurogenesis to blood brain barrier biology. Dysregulated Wnt signalling can thus have significant consequences for normal brain function, which is becoming increasingly clear in Alzheimer’s disease (AD), an age-related neurodegenerative disorder that is the most prevalent form of dementia. AD exhibits a range of pathophysiological manifestations including aberrant amyloid precursor protein processing, tau pathology, synapse loss, neuroinflammation and blood brain barrier breakdown, which have been associated to a greater or lesser degree with abnormal Wnt signalling. Here we provide a comprehensive overview of the role of Wnt signalling in the CNS, and the research that implicates dysregulated Wnt signalling in the ageing brain and in AD pathogenesis. We also discuss the opportunities for therapeutic intervention in AD via modulation of the Wnt signalling pathway, and highlight some of the challenges and the gaps in our current understanding that need to be met to enable that goal.

## 1. Introduction

The term dementia encompasses a group of devastating disorders with characteristic declines in cognition, function and behaviour, which eventually severely impair patients’ ability to perform instrumental and/or basic activities [1,2]. Beyond the patient, this profoundly impacts caregivers, families and society as a whole, which is facing ever increasing health care costs associated with dementia [3]. Alzheimer’s disease (AD) is by far the most common form of dementia accounting for 60–80% of cases [3], with an estimated 40 million diagnosed patients worldwide [4]. Current therapeutic options are restricted to symptomatic treatments such as cholinesterase inhibitors or the glutamate receptor antagonist memantine, which have a limited effect on memory and cognition [5]. As disease-modifying therapies, which target the disease process, are not yet available [6,7], AD dementia inevitably results in death within 5–12 years of symptom onset [8].

While the need for disease-modifying therapies that prevent disease onset or slow disease progression is clear, the lack of such treatments reflects the complicated, multifactorial pathobiology of AD. Patients often present with a range of disease manifestations, but what uniquely defines AD is the presence of protein aggregates in the form of extracellular deposits of β-amyloid (Aβ) as diffuse and neuritic plaques, as well as intracellular neurofibrillary tangles (NFTs) consisting of hyperphosphorylated tau [9] observed in post-mortem tissue. Yet, as of writing, therapeutic approaches targeting these two disease hallmarks have not proven to be effective in clinical trials [6,7] (although approaches targeting tau are at a relatively early stage of evaluation in patients [10]). This is likely to be for a number of reasons, beyond the scope of this review. However, it does raise the possibility that an effective therapeutic strategy will require the targeting of several pathological manifestations via different biological pathways, and that this may change over the time course of the disease.

The Wnt signalling system plays a crucial role in many cellular processes such as cell differentiation, migration, and tissue homeostasis. In the central nervous system (CNS), Wnt signalling regulates developmental programmes and, as is increasingly recognised, it modulates a number of aspects of the mature brain such as synapse number and function, the integrity and function of the blood brain barrier (BBB), as well as the biology of microglia (the resident immune cells of the CNS). These key facets of the mature brain are significantly impacted in AD, raising the possibility that dysregulation of Wnt signalling may play an important role in a number of different aspects of this complex disease [11,12].

In this review, we aim to provide a systematic overview of the role of Wnt signalling in the CNS, together with what is known regarding dysregulation of this signalling system and how this contributes to certain pathological processes in AD. We also consider potential opportunities for new therapeutic approaches in AD through selective modulation of Wnt signalling, while recognising some key gaps in our knowledge that need to be addressed.

## 2. Wnt Signalling Pathways—An Overview

Wnt ligands are lipid-modified [13], secreted glycoproteins, which, upon binding to cell surface receptors, trigger intracellular signalling pathways that regulate various biological processes such as the cell cycle, cell migration and establishment of cell polarity [14,15]. The genomes of most mammals harbour 19 Wnt genes that can be grouped into 12 conserved subfamilies. These ligands are recognised by a heterodimeric receptor complex on the cell surface comprising Frizzled (Fz) as well as single-pass transmembrane co-receptors LRP5, LRP6, Ror1, Ror2, or Ryk proteins (Figure 1) [16,17,18,19]. The mammalian genome harbours 10 Fz genes, which encode 7-transmembrane (7TM) receptors that exhibit an N-terminal, extracellular, large cysteine-rich domain (CRD) used for Wnt binding [20,21,22]. However, ligand-receptor interactions are promiscuous as evidenced by the fact that there are multiple, non-mutually exclusive ligand receptor combinations [23,24].

Wnt ligand binding by the receptor complex induces conformational changes of the receptor complex and phosphorylation of target proteins as intracellular signalling pathways are initiated. These signalling pathways can broadly be classified into the canonical Wnt/β-catenin and non-canonical β-catenin-independent signalling cascades. The activation of specific pathways depends on, among various other factors, the exact identity of involved ligand and receptor isoforms, the expression of which is under tight spatiotemporal control [23].

The canonical Wnt/β-catenin signalling cascade is initiated upon binding of Wnt to the Fz-LRP receptor complex (Figure 1a,b) [25]. This subsequently leads to the inactivation of a multiprotein complex consisting of CK1α, GSK-3β, Axin, and APC, which usually phosphorylates β-catenin, thus marking it for proteasomal degradation [26]. Accumulating β-catenin can now translocate to the nucleus where it associates with transcription factors from the TCF/Lef family [27,28] to activate transcription of Wnt/β-catenin target genes with known roles in proliferation, fate specification and differentiation in development as well as adult tissue homeostasis [29].

Initiation of the non-canonical Wnt/PCP (planar cell polarity) cascade requires Wnt binding to a Fz-Ror/Ryk receptor complex (Figure 1c) [30,31,32]. Intracellularly, this causes the activation of small GTPases RhoA, Rac, and Cdc42 [33,34], which subsequently activate the downstream kinases JNK and ROCK [35,36], which in turn regulate actin and microtubule cytoskeletons. As such, the Wnt/PCP cascade plays a vital role in the control of cell/tissue polarity and cell migration [37].

The Wnt/Ca^2+^ cascade requires binding of Wnt ligand to Fz receptor, which intracellularly triggers G-protein coupled signalling [38]. This in turn activates phospholipase C (PLC) [39], which stimulates the release of Ca^2+^ from intracellular stores via the signalling molecule inositol triphosphate (IP3) [40]. The mobilised Ca^2+^ then stimulates the Ca^2+^-sensitive protein kinases protein kinase C (PKC) [41] and Ca^2+^/Calmodulin-dependent protein kinase II (CaMKII) [42], as well as the Ca^2+^ sensitive transcription factor NF-AT. Through these effector proteins the Wnt/Ca^2+^ cascade regulates many processes, ranging from developmental cell fate determination, cell/tissue migration, cell differentiation, and inflammatory response mediation [43].

Wnt signalling can furthermore be modulated by a number of endogenous agonists and antagonists [44], which are important for the fine-tuning of Wnt signalling-regulated processes. There are seven secreted antagonist families (the Dickkopf proteins (Dkks), secreted Frizzled-related proteins (sFRPs), Wnt inhibitory factor 1 (WIF-1), Wise/SOST, Cerberus, insulin-like growth factor binding protein 4 (IGFBP-4), Notum and four transmembrane Wnt antagonist families (Shisa, Wnt-activated inhibitory factor 1 (Waif1/5T4), adenomatosis polyposis coli down-regulated 1 (APCDD1), and Tiki1). They exert their function either by sequestering/inactivating secreted Wnt (e.g., WIF1, Cerberus, sFRP, Notum) or by blocking/sequestering elements of receptor complexes (e.g., Dkk1, Wise/SOST, IGFBP-4). For example, Dkk1 sequesters LRP6, thus preventing its heterodimerisation with Fz8 to block canonical Wnt/β-catenin signalling [45]. When the Dkk1 co-receptor Kremen 2 (Krm2) is present, this additionally leads to the endocytosis of LRP5/6-Krm2-Dkk1 complexes [46]. The related Dkk2, however, can act both as an activator and as an inhibitor of the Wnt/β-catenin cascade [47], in the presence Krm2, Dkk2 functions as an LRP6 antagonist, while in its absence it functions as an activator.

Conversely, there are two families of secreted proteins that act purely as Wnt agonists: R-spondins (Rspo) and Norrin. R-spondins stimulate canonical Wnt/β-catenin signalling by promoting the internalisation of the transmembrane E3 ubiquitin ligase ZNRF3 that usually marks Fz and LRP6 for degradation [48]; this is mediated by the Rspo receptors Lgr4, Lgr5, and Lgr6 [49]. R-spondins can also stimulate Wnt/PCP signalling via a mechanism that requires Wnt5a-Fz7 signalling, Rspo3 binding to the four transmembrane proteoglycan syndecan 4, and syndecan 4-dependent endocytosis of the entire Wnt5a-Fz7-Rspo3-syndecan 4 complex [50]; in contrast to the action of Dkk1/Krm2 on canonical Wnt/β-catenin signalling through Fz8/LRP6, this internalisation was shown to be crucial for Wnt/PCP signal transduction. Finally, Norrin, although structurally unrelated to Wnts, binds Fz4/LRP5 to activate Wnt/β-catenin signalling [51].

The Wnt cascades thus constitute a highly complex signalling network with a range of different functions and roles in development, mature homeostasis, ageing, as well as disease. The remainder of this review will provide more detailed insight into Wnt signalling in the brain with a focus on the deregulation of Wnt signalling in AD.

## 3. Wnt Signalling in the Brain

Wnt signalling plays an important role in various aspects of the brain, ranging from brain development to normal brain function. Indeed, the gene expression of various Wnt and Fz receptor isoforms is subject to tight spatio-temporal control, as has been reviewed in [52]. Unsurprisingly, altered Wnt signalling strength can have detrimental effects on the brain, as is for instance observed during ageing where reduced Wnt signalling is evident. In this section we summarise the importance of Wnt cascades within the healthy brain lifespan.

### 3.1. Wnt Signalling during Brain Development

Wnt signalling plays an important role during various aspects of brain development. As secreted morphogens, Wnts as well as modulators of Wnt signalling, regulate the establishment of distinct tissue domains within the CNS along Wnt concentration gradients leading to the specification of the brain as a whole and substructures within the brain in a process that is more generally termed tissue patterning. Important Wnt signalling components in this process include Wnt-1, Wnt3a, Wnt8c, Dkk1, sFRPs, and Notum [53,54,55,56,57,58,59]. Fittingly in the context of AD, Wnt signalling is also required for the development of brain structures including the hippocampus among others, which is integral for memory [60]. There, Wnt3a regulates both neuronal progenitor cell (NPC) expansion and neuronal differentiation via canonical signalling [61,62,63,64,65]. Wnt3a is further accompanied by Wnt5a and Wnt7b, which respectively regulate axonal and dendritic differentiation (axon specification, axon growth, and dendritic tree formation) via non-canonical Wnt/PCP signalling [66,67,68,69,70]. One further neuronal differentiation aspect also regulated by Wnt signalling is synaptogenesis, as has been reviewed in [52]. Canonical Wnt signalling (via Wnt7a, Wnt7b, and Wnt3a) and non-canonical Wnt signalling (via Wnt5a) respectively increased and decreased pre-synaptic assembly in rat primary hippocampal neurons [71,72,73,74,75], indicating that canonical and non-canonical Wnt pathways need to be well balanced to ensure adequate synapse numbers (this concept is relevant in the context of AD, as will be described in Section 4.3). Conversely, post-synaptic assembly is driven by Wnt5a-mediated non-canonical Wnt/PCP as well as Wnt7a-mediated Wnt/Ca^2+^ pathways, at least in rat primary hippocampal neurons [76,77,78,79]. Taken together, Wnt signalling is highly important in the brain at many stages of development.

### 3.2. Wnt Signalling in the Mature Brain

Beyond the developmental period, Wnt signalling remains integral to brain function in adulthood, especially at the level of the synapse. Wnts and Wnt regulatory proteins are released locally at the synapse. This ensures homeostatic control over synaptic connectivity even during basal synaptic transmission. A study on mature cultured rat hippocampal neurons and adult mouse brain slices indicates the presence of an endogenous, non-canonical Wnt5a signalling tone that regulates excitatory synapse activity [76]; blocking endogenous Wnt signalling by exogenous sFRP antagonist treatment reduced the amplitudes of evoked field excitatory post-synaptic potentials (fEPSPs) and miniature excitatory post-synaptic currents (mEPSCs) according to electrophysiology recordings. In mature mouse brain slices containing the entorhinal cortex-hippocampal circuit, antibodies against canonical ligands (Wnt3a and Wnt7a) and non-canonical ligand (Wnt5a), as well as application of sFRP-2 reduced circuit activity; this led the authors to conclude that constitutive Wnt release contributes to network maintenance, presumably by affecting synaptic mechanisms [80]. Among the two glutamate receptors present at the post-synaptic terminus of excitatory synapses, α-amino-3-hydroxy-5-methyl-4-isoxazolepropionic acid receptor (AMPAR) and N-methyl-D-aspartate receptor (NMDAR), Wnt5a scavenging with sFRP treatment specifically reduced basal NMDAR, but not AMPAR, currents (Wnt5a usually increases NMDAR currents in a Ca^2+^, PKC, and JNK dependent manner) [81]. Another sFRP treatment study revealed that Wnt signalling regulates basal synaptic transmission by maintaining pre-synaptic termini in the CA3 stratum lucidum of the hippocampus, which receives axonal inputs from the dentate gyrus (DG) area of the mouse hippocampus [82]. Taken together, these studies suggest that a basal endogenous Wnt signalling tone is required to ensure structural and functional synaptic connectivity.

Beyond basal synaptic transmission, Wnt signalling pathways also refine synaptic contacts (strengthening or weakening) in an environment and activity-dependent manner—a mechanism known as synaptic plasticity [83]. In cultured rodent hippocampal neurons and slices, Wnt7a increased the density and maturity of dendritic spines, and further enhanced synaptic strength by AMPAR recruitment to synaptic spines in a CaMKII-dependent manner [79,84], while simultaneously enhancing neurotransmitter release at CA3-CA1 synapses [72,85]. Protocols that elicit long term potentiation (LTP; the sustained strengthening of a synapse thought to be the cellular model of learning and memory), stimulate Wnt3a synaptic secretion upon post-synaptic NMDAR activation in mice [86]. Newly released Wnt3a subsequently binds synaptic Fz receptors to canonically initiate the expression of LTP target genes. Electrophysiologically, canonical Wnt3a enhanced the LTP response by increasing fEPSPs in adult mouse hippocampal slices [86]. At least in rat and mouse hippocampal slices, this appears to be complemented by non-canonical Wnt5a signalling, which, via NMDAR current potentiation, facilitated the induction of LTP [81,87]. The above findings were also supported by an in vivo study, which demonstrated that environmental enrichment increased Wnt7a/b levels in the mouse hippocampus; this in turn induced synaptic remodelling in the CA3 of the mouse hippocampus, suggesting that Wnt signalling can bring about structural changes even in the adult brain [82]. Furthermore, spatial information storage was also correlated with increased *Wnt7* and *Wnt5a* expression in the mouse DG 7 days and 30 days after water maze training [88]. Indeed administration of activators of canonical (WASP-1) and non-canonical signalling (FOXY-5) improved episodic memory in a water maze spatial task [87]. A subsequent study provided further proof that canonical Wnt signalling is necessary for hippocampal memory consolidation in mice, as canonical Wnt signalling was enhanced following object recognition training, which could be blocked by hippocampal injection of Dkk1 [89].

Neurogenesis, which in the adult brain is restricted to a select few brain regions including the subgranular zone (SGZ) of the DG, is also modulated by Wnt signalling. Neurogenesis has been reported to influence memory [90], though how exactly adult neurogenesis contributes to memory and whether it even exists in humans is still a matter of ongoing debate [91]. Nevertheless, research in rats has shown that Wnt3-mediated canonical signalling stimulates adult neurogenesis in the DG in vitro and in vivo, and that loss and gain of function approaches to block or stimulate Wnt signalling in the DG respectively attenuated and increased adult neurogenesis [92,93]. Furthermore, blocking Wnt signalling in the DG of adult rats impaired spatial and object recognition memory [94]. Wnt3a canonical signalling was shown to transcriptionally induce expression of *NeuroD1* in mice, which has a role in the neuronal differentiation of DG-resident granule cell neurons [95,96]. In addition to Wnt3, canonical Wnt7a signalling may also play a role in adult neurogenesis in the mouse DG, where a decreased NPC population, reduced neurogenesis, and impaired granule cell neuronal differentiation were evident in *Wnt7a* knockout mice compared to control; these results indicated that Wnt7a regulates both NPC proliferation (via cyclin D1) and neuronal differentiation (via Neurog2) [97]. The endogenous Wnt inhibitor sFRP3 also regulates both NPC proliferation and neuronal differentiation, where adult *sFrp3* knockout mice exhibit both increased NPC proliferation, neurogenesis, and accelerated dendritic growth and spine formation of new-born neurons [93]. A similar negative regulatory function was recently described for endogenous Wnt antagonist *Notum*, which limits adult olfactory bulb neurogenesis from the subventricular zone in mice [98]. Taken together, Wnt signalling is crucial for the regulation of neuronal circuits, both at a basal level as well as during activity-related plastic changes that likely contribute to memory and learning.

### 3.3. Wnt Signalling in the Ageing Brain

Ageing is a multifactorial process of progressive detrimental changes occurring at the molecular, cellular, tissue, and organ level and is associated with functional decline [99]. This also applies to the ageing brain, where impairments in cognitive function and memory are evident. A body of work suggests that this might be due reductions in both synapse function and adult neurogenesis, particularly in the hippocampus [100,101,102,103]. Interestingly, dysregulated Wnt signalling is also observed in older age. Indeed, several genes involved in Wnt signalling are downregulated in the brains of advanced age rodents, pointing towards reduced Wnt signalling tone. In 36 months old mice, the Wnt ligands *Wnt2*, *Wnt4*, and *Wnt8a*, as well as the canonical transcription factors *Lef1*, *Tcf3*, and the transcriptional output *Ccnd2* were significantly decreased compared with 5 month controls, as determined by quantitative PCR [104]. Meanwhile, astrocytic *Wnt3a* expression and secretion in the DG progressively declined, while *Dkk1* expression increased with age [105,106]. In a comparison between 12 or 24 month old rats with 4 month old controls, reduced total as well as nuclear β-catenin protein were detected in the hippocampus alongside increased phosphorylated (active) GSK-3β, indicative of diminished canonical Wnt signalling in aged animals [107]. The aged human brain also exhibits gene expression patterns that are in line with reduced Wnt signalling, specifically a downregulation of the Wnt ligands *WNT2B*, *WNT6*, *WNT7a*, and Fz receptors *FZD2* and *FZD3*, alongside an upregulation of *GSK3B* and *SFRP1* in the prefrontal cortex [108]. Further evidence implies that menopause-related estrogen deprivation increases *DKK1* expression in the mammalian hippocampus [109,110].

Given the well documented role that Wnt signalling plays in the healthy, mature brain many studies now suggest that the Wnt signalling-dampened aged brain is prone to reductions in both synapse function and adult neurogenesis, thus contributing to cognitive decline. Indeed, conditional *Dkk1* knock out in NPCs restored adult neurogenesis in old age mice, and increased dendritic complexity as well as neuronal activity in the DG [106]. This concomitantly restored working memory and memory consolidation in behavioural tests. Conditional *Dkk1* overexpression in mature 3-6 months old mice (thus modelling *Dkk1* upregulation at more advanced ages) had profound consequences on synapses, resulting in excitatory synapse loss, reduced LTP, and learning/memory deficits [111]. Crucially, this was fully reversible upon cessation of *Dkk1* overexpression. In another study, age-related decline in astrocytic *Wnt3a* expression and secretion reduced adult neurogenesis in mice; however, this could be reversed by overexpressing *Wnt3a* in astrocytes or by stimulating endogenous astrocytic Wnt3a production with exercise [105]. Finally, stimulation of canonical and non-canonical Wnt signalling with small molecule “activators” FOXY-5 and WASP-1, improved excitatory synapse transmission, LTP and episodic memory in mature mice [87].

The underlying tenor is therefore that Wnt signalling strength declines with increasing age and that this can lead to cognitive decline. Equally, restoring deficient Wnt signalling may be able to reverse age-related cognitive deficits, although the strategy of how this can be effectively achieved in humans remain to be elucidated (see Section 4.6). As discussed in Section 5, any future ambitions to enhance Wnt signalling using small molecules or gene therapy approaches requires careful consideration with respect to the exact Wnt pathway target, cellular specificity, and dose so as to avoid unwanted side effects including tumour formation.

## 4. Wnt Signalling in AD

Given the connection between age-related cognitive decline and decreased Wnt signalling strength (discussed in the previous section), it is perhaps unsurprising that Wnt signalling pathways are also suppressed in AD given the strong connection between AD and its biggest risk factor–age [112,113]. Indeed, the majority of AD cases are detected at an advanced age (sporadic late onset AD or LOAD) supporting the fact the AD is predominantly an age-related disorder of the brain. However, mutations in certain genes result in early onset heritable/familial AD or FAD; see Section 4.1). While FAD is associated with mutations in amyloid precursor protein (*APP*) or APP processing genes (*PSEN1*, *PSEN2*, encoding components of the γ-secretase complex), numerous additional risk genes (including *APOE*, *TREM2*, and *UNC5C*) as well as environmental factors confer susceptibility to LOAD. Genetic studies have been invaluable in shedding light on the multiple pathological mechanisms observed in AD, as will be discussed below. In several cases where gene variants have been linked to AD, these genes also show connections to Wnt signalling pathways.

Among the first pieces of evidence indicating that abnormal Wnt signalling could be involved in AD came from a study that discovered a polymorphism in the *GSK3β* promoter, which increased its activity and conferred increased susceptibility to LOAD [114] (given the known biological role of GSK-3β, increased activity reduces canonical Wnt signalling by promoting β-catenin degradation). Further supporting the link between reduced canonical signalling and AD, a single nucleotide polymorphism (Ile-1062 → Val) and a novel splice variant within the canonical Fz co-receptor encoding gene *LRP6* were also associated with LOAD; functionally, in HEK293T cells, both reduced canonical Wnt signalling [115,116]. In addition, *LRP6* mRNA and protein levels were significantly decreased in human AD brains compared to controls [117] (for an extensive review on the connection between *LRP6* and AD the reader is referred to [118]). Another Wnt signalling suppressing component in AD is the induced neuronal expression of the endogenous Wnt signalling antagonist *DKK1* in the brains of post-mortem AD patients as well as in AD mouse models [119,120]. In fact, it was demonstrated that AD-associated Aβ fibrils induce *Dkk1* expression in acute mouse hippocampal slices within several hours [121]. Importantly, DKK1 is a ligand for LRP6 and exerts it action by removing LRP6 from functional canonical Fz-LRP6 receptor heterodimers [45]. Interestingly, the *Dkk1* homologue *Dkk2* was also significantly upregulated in various AD mouse models, specifically within the myeloid cell lineage that includes microglia and other immune cells [122]. The protein clusterin also participates in the Aβ-DKK1 pathway, as discussed below in Section 4.3. From a purely genetic standpoint, the encoding gene *CLU* had previously been identified as a major LOAD risk gene in genome wide association studies (GWAS), where various AD-linked single cell nucleotide polymorphisms were found to be associated with AD [123,124].

ApoE is a regulator of lipid homeostasis predominantly produced by astrocytes within the CNS. By binding to ApoE receptors on neurons, it transports cholesterol to neurons, which lack cholesterol producing capabilities. [125,126]. However, among the three existing polymorphic alleles *ε2*, *ε3* and *ε4*, the *APOE ε4* allele is found in ~40% of AD cases despite only being represented in 13.7% of the general population, thus classifying it as a major AD risk gene [127]. The nature of the role of ApoE in AD is multifaceted and still not fully understood, but highlighted by increased Aβ binding and deposition in *APOE ε4* carriers [128]. In addition, collective evidence suggests that ApoE also interacts with Wnt signalling pathways in an AD-relevant manner: ApoE4-mediated Aβ pathology in the *APP/PS1* AD mouse model required the expression of its neuronal receptor *LRP1* [129], which had previously been found to suppress Wnt3a-driven canonical Wnt signalling in HEK cells by interacting with Fz1 [130]. Furthermore, ApoE4 treatment in PC12 cells, more so than ApoE2 and ApoE3, suppressed Wnt7a-stimulated canonical signalling [131].

Genetic variants of triggering receptor expressed on myeloid cells-2 (*TREM2*) have also been linked to increased susceptibility to LOAD [132,133]. Although relatively rare overall, carriers of the most common and best studied *TREM2* variant (R47H) had a 2–4 fold increased risk for LOAD, which is in the same range as the *APOE ε4* allele (minor allele frequency is population dependent but reaches up to 0.63% in the Icelandic population [132]). Within the CNS, *TREM2* is predominantly expressed by microglia where it was shown to modulate canonical Wnt signalling to support microglial survival and microgliosis, both of which are markedly impaired in *Trem2^−/−^* mice [134]. These and other findings highlight the importance of the CNS immune system in AD, the proper function of which requires canonical Wnt signalling (this will be expanded upon in Section 4.4).

In addition to genetic analyses of AD pathogenic mechanisms, dynamic gene expression changes associated with AD and the resulting protein level changes are equally important for enabling our understanding of this disease. In a recent mass spectrometry based proteomics study, protein level changes were assessed in the brains and cerebrospinal fluid of AD patients and compared with control, prodromal, and mild cognitive impairment (MCI) cases [135]. Indeed, Wnt signalling related proteins were among those proteins exhibiting increased levels. These proteins included the Wnt ligands *WNT5A*, *WNT5B* as well as the endogenous antagonists *SFRP1* and *FRZB* (SFRP3). While elevated SFRP1 and FRZB levels would be consistent with a reduced Wnt signalling tone, increased presence of WNT5A and WNT5B may reflect a pathological shift from canonical to non-canonical signalling that reduces synapse stability due to increased actin cytoskeleton dynamics [136] (see Section 4.3). In further support of a reduced Wnt signalling tone in AD brains, the phosphoproteome (phosphorylation status of proteins within the proteome) revealed significant phosphorylation increases of GSK-3β target proteins, which is indicative of increased GSK-3β activity and reduced canonical Wnt signalling strength [135]. These findings are corroborated by a further proteomics study, which reported that canonical Wnt signalling was dysregulated in specific human AD brain regions versus control [137]. The hippocampus for instance exhibited 17 differentially expressed Wnt signalling related proteins, including GSK-3β, GSK-3α, and AKT3, which were all downregulated, as well as the DKK homologue DKK3, which was upregulated. While the role of DKK3 as an agonist or antagonist of Wnt signalling is context dependent [138,139], the downregulation of GSK-3β, GSK-3α, and AKT3 would be expected to result in increased β-catenin stability and hence canonical Wnt signalling. This would be incompatible with the widespread notion that a reduction of canonical Wnt signalling contributes to AD pathology.

These findings notwithstanding, Wnt signalling dysregulation has been found to be a prominent feature in AD and further studies are needed to fully understand all the aspects involved. The following sections will provide a more detailed description of the role of Wnt signalling in the various pathobiological disease manifestations of AD (schematically summarised in Figure 2).

### 4.1. Wnt Signalling, APP Processing, and AD

One of the defining hallmarks of AD is the extracellular aggregation of Aβ and the formation of plaques. Its pathologic accumulation in the brains of AD patients is the result of aberrant proteolytic processing of the membrane–embedded Amyloid Precursor Protein (APP; APP family members usually play an important role during brain development and maintenance) (reviewed by [140]). Whereas the non-amyloidogenic processing pathway requires successive cleavage by α- and γ-secretase, the amyloidogenic pathway (the pathway resulting in the cleaved 39–42 amino acid long Aβ peptide (Aβ) relevant to AD) requires cleavage by β- and γ-secretases [141]. It is the pathological accumulation of Aβ caused by increased amyloidogenic processing and/or impaired Aβ clearance that contributes to AD-related formation of Aβ plaques. While the 40 and 42 amino acid long peptides Aβ40 and Aβ42 are most prevalent in AD, it is Aβ42 that is particularly neurotoxic [141]. In keeping, mutations in genes involved in APP processing often lead to FAD. In fact, mutations in *PSEN1* (presenilin 1 or PS1), encoding the catalytic subunit of γ-secretase, are the most common cause of FAD, followed by mutations in *PSEN2* and *APP* [142,143,144,145].

The link between APP processing and impaired Wnt signalling can be exemplified by the fact that *APP* locus duplication causes FAD, involving a mechanism whereby APP binds β-catenin–as demonstrated in vitro, in vivo and in AD patients–preventing its nuclear translocation and thus canonical Wnt signalling [144,146]. β-catenin was also shown to interact with PS1; however, while this interaction usually stabilised β-catenin in cell lines, FAD-mutant PSEN1 exhibited reduced β-catenin stability and protein levels [147] and impaired trafficking to the nucleus [148] in HEK cells. This suggests that FAD-associated *PSEN1* mutations negatively regulate canonical Wnt signalling. However, conflicting findings have been reported where human *PSEN1* mutations increased steady state levels of β-catenin in neuronal (PC-12) and fibroblast cell lines (EcR293) [149,150]. While this is difficult to reconcile, it was suggested that these different outcomes may have been observed due to the use of different experimental models, different time points, the transient nature of the Wnt pathway involving negative feedback loops, measuring cytoplasmic rather than nuclear β-catenin, or potential cross-talk with the Notch pathway (the Notch receptor is a γ-secretase substrate and can context-dependently antagonise Wnt signalling by modulating β-catenin stability) (reviewed by [151]). Nonetheless, it remains highly likely that FAD-*PSEN1* mutations, and by extension Aβ, are linked to dysregulated canonical Wnt signalling via dysregulation of β-catenin. Of note, reduced β-catenin levels were not observed in sporadic AD patients [147].

Thus, common FAD-associated variants of genes involved in APP processing can negatively impact Wnt signalling. However, the inverse is also true. Wnt signalling influences APP processing where inhibition of Wnt signalling was shown to induce amyloidogenic APP processing and Aβ aggregation while Wnt3a stimulation decreased the expression and activity of the amyloidogenic β-secretase *BACE1* and amyloidogenesis in vivo and in vitro [152,153,154]. In further support, the canonical Wnt co-receptor *LRP6* was found to be downregulated in human AD brains, which was correlated with a reduction of β-catenin and an increase of Aβ42 [117]. The same study then confirmed a causative relationship by showing that *Lrp6* loss of function increased Aβ40 and Aβ42 production in neuronal cell lines, in the *APP/PS1* AD mouse model, and also in wild type mice. Mechanistically, based on experiments in HEK293 cells, it was suggested that LRP6 directly binds to APP, leading to an increased cell surface expression of APP favouring non-amyloidogenic processing and decreased Aβ production [117]. Consistent with the above study, chronic administration of small molecule antagonists of canonical Wnt signalling (ICG001 or XAV939) into wild type mice or the *J20* AD mouse model also increased amyloidogenic APP processing [155]. Together, these two studies highlight the importance of Wnt signalling for correct APP processing, especially given the fact that Wnt signalling attenuation triggered AD-like amyloidogenic changes (as well as memory deficits) including in wild type mice. Overall, it appears that APP processing and Wnt signalling are engaged in a cross-regulatory network, which deteriorates into a pathologic feedback loop in AD as Wnt signalling is depressed, causing increased amyloidogenic processing, which in turn further depresses Wnt signalling [117]. Consistent with this hypothesis, feedback loop intervention by activation of Wnt signalling with LiCl (a GSK-3β inhibitor), Wnt3a or Wnt7a rescued the neurodegeneration and behavioural impairments induced by Aβ fibrils [156,157,158], while Aβ sequestration restored canonical Wnt signalling [159] in rat primary neurons as well as in cell lines.

### 4.2. Wnt Signalling, Tau Pathology, and AD

The second major hallmark of AD is the emergence of tau pathology, otherwise known as tauopathy, in the shape of neurofibrillary tangles (NFTs) within neurons. NFTs are made up of the microtubule associated tau protein in a hyperphosphorylated state (tau usually has a microtubule stabilising function, which is governed to a large extent by its phosphorylation status that is subject to the action of kinases and phosphatases; [160,161]). Multiple neuronal processes critically depend on proper tau function, including mitochondrial transport, synaptic transmission, and autophagy (reviewed in [162]). Consistent with its important role, mutations in tau-encoding gene *MAPT* frequently negatively impact microtubule dependent activities especially within neurons, due to the emergence of tau hyperphosphorylation and tauopathy. However, unlike mutations in *APP* and APP processing genes, mutations in *MAPT* typically result in in a different type of dementia called frontotemporal dementia (FTD) rather than FAD [163,164]. Nonetheless, tau is inextricably linked to AD as evidenced by the presence of important mechanistic and perhaps synergistic linkages between Aβ and tau but the precise nature of this relationship is complex. For instance, tau pathology may arise independently of cortical Aβ, but equally, tau spreading and cognitive decline are facilitated by the presence of cortical Aβ according to studies on various AD mouse models [165,166,167,168,169,170,171,172].

In addition to phosphorylating and thereby inhibiting β-catenin, GSK-3β is also one of the major kinases responsible for tau phosphorylation [173,174]. Canonical Wnt signalling usually inactivates GSK-3β. However, as Wnt signalling strength is reduced in AD, the activity of GSK-3β is increased, thus contributing to AD-related tau hyperphosphorylation [108,175,176,177,178,179]. As discussed above, Wnt signalling offers pathological opportunities for crosstalk between amyloidogenic APP processing and tau hyperphosphorylation. Accordingly, FAD-associated mutations in the APP processing related gene *PSEN1* (known to reduce Wnt signalling; see Section 4.1), increase GSK-3β activation and tau phosphorylation in human brain biopsies, an effect that is likely related to the ability of PS1 to associate with both GSK-3β and tau [180]. Furthermore, studies have confirmed that Aβ treatment activates GSK-3β and increases tau hyperphosphorylation in cultured neurons, acute brain slices, as well as in vivo, which could be mimicked by Dkk1 treatment and blocked with the GSK-3β inhibitor 6-bromo-indirubin-3′-oxime [175,176]. There was also a concomitant decrease in the activity of the tau phosphatase PP2A in rat hippocampal slices, indicating that the usually tightly controlled balance between tau phosphorylation and dephosphorylation is now more generally shifted towards phosphorylation ultimately resulting in tau hyperphosphorylation and tauopathy [175].

Another link between tau and Wnt signalling was recently described, where it was discovered that tau itself can acetylate β-catenin, which increased β-catenin stability and nuclear translocation in HEK cells and rat primary neurons, and further promoted cell survival [181,182]. However, in hippocampal extracts from AD patients, the levels of β-catenin and K49-acetylated β-catenin were increased and further positively correlated with phosphorylation levels of tau at S199 [182]. While decreased β-catenin levels, and hence Wnt signalling, would be more intuitive in the context of AD, it should be pointed out that no distinction between nuclear and total β-catenin levels was made in the above study. Thus, even though a more nuanced evaluation of β-catenin would be required (see also relevant discussion in Section 4.1), β-catenin levels were also increased in *PSEN1* related human FAD brains [149,150], indicating that dysregulation of canonical Wnt signalling in AD may manifest via either decreased or increased β-catenin. The authors of [182] speculated whether accumulated tau could help neurons escape acute apoptosis while simultaneously causing chronic neurodegeneration.

### 4.3. Wnt Signalling, Synapses, and AD

The pathophysiological processes that lead to AD start many years before the diagnosis of AD dementia [6]. During this long preclinical phase, synaptic dysfunction, as determined by ^18^F fluorodeoxyglucose positron emission tomography or functional magnetic resonance imaging, appears to occur relatively early and before brain atrophy, abnormal cognition, and decreased clinical function become detectable [183]; however, it is important to highlight that the exact timeline with respect to the onset of the individual clinical and pathological hallmarks is still subject to refinement as detection and screening tools are continually improved (e.g., [184]). Nonetheless, it is generally accepted that Aβ pathology emerges before synapse dysfunction. This relationship is not only temporal but also aetiological in nature as Aβ contributes to impaired synaptic function and synapse loss by acting as a synaptotoxin [185,186,187,188,189,190]. Given the fact that synapses are crucial for cognitive function and that synapse loss correlates with the degree of cognitive impairment in AD patients, understanding the molecular mechanisms that link Aβ, synapse loss and cognitive decline is thus of utmost importance [191,192,193,194].

It has emerged over the years that the endogenous antagonist of canonical Wnt signalling Dkk1 plays a central role in this process. In cultured rat neurons, Aβ oligomer treatment rapidly induced *Clu* (clusterin) expression and reduced its protein secretion, which resulted in p53-dependent induction of *Dkk1* [195]. Of note, DKK1 expression is also evident in neurons of AD patients [119]. This pathway appears to be at least partially tied to synapses since clusterin was demonstrated to accumulate within synapses in human AD brains [196]. As was shown in a study on acute adult mouse hippocampal slices, *Dkk1* induction occurs rapidly, where Aβ oligomers induced *Dkk1* expression in CA1, CA3 and DG at mRNA and protein levels within 3 h of treatment [121]. Furthermore, in mature rat hippocampal neuron cultures with established synapses, Dkk1 treatment rapidly reduced the number of synaptic sites, both at the pre- and post-synaptic level based on immunocytochemistry and electron microscopy data. This was recapitulated in vivo, where inducible *Dkk1* expression resulted in reduced synaptic transmission, impaired LTP, enhanced long term depression (LTD), synapse degeneration and memory deficits in the hippocampus and striatum [111,197]. While LRP5/6 are Fz co-receptors for Wnt ligands, LRP6-Krm1 heterodimers are also the main receptor for Dkk1 and thus Dkk1 acts by removing functional Fz-LRP6 co-receptors for canonical Wnt ligands [45,198,199,200]. Common *LRP6* variants are associated with late onset AD (see Section 4 and Section 4.1) while conditional *Lrp6* loss of function in neurons elicited synaptic impairment in aged mice and exacerbated amyloid pathology in the *APP/PS1* AD mouse model [115,117]. Equally, silencing *Krm1* with *miR-431* prevented Aβ-related synapse loss in cultured neurons obtained from the *3xTg* AD mouse model [200]. Thus, Dkk1-mediated synapse disassembly is mediated by the attenuation of canonical Wnt signalling [111,136,195,201].

Crucially, synapse loss as a response to Dkk1 exposure was reversible in vitro and in vivo, where synapse number and function was restored following removal of Dkk1 by washout or by cessation of *Dkk1* over-expression [111,121], indicating that the normalisation of Wnt signalling may enable restoration of synapses in AD patients. In fact, activation of both canonical and non-canonical pathways were proven to counteract the synaptotoxic effects of Aβ: a potentiator of Wnt3a-mediated canonical signalling, WASP-1, as well as the non-canonical ligand Wnt5a both rescued hippocampal synaptic impairments triggered by Aβ oligomers [154,190]. However, there is now also mounting evidence suggesting that activation of non-canonical PCP signalling may exacerbate Aβ oligomer synaptotoxicity. Indeed, Aβ-mediated *Dkk1* induction activated the Wnt/PCP pathway via Fz-Vangl2 co-receptors and the resulting modulation of cytoskeletal dynamics contributed to synapse loss, which could be reversed in rat primary cortical neurons by blocking this pathway using inhibitors of the actin cytoskeleton regulatory protein ROCK [111,136,195,201]. The mechanism of action of Dkk1 may thus involve both the reduction of canonical Wnt signalling as well as the activation of the Wnt/PCP pathway, both of which may result in the destabilisation and eventual loss of synapses [136,201]. Adding a further layer of complexity, APP physically interacted with LRP6 and Vangl2 and co-activated canonical and Wnt/PCP signalling respectively in cell line reporter assays. However, over-expression of an FAD-related *APPSwe* variant was less able to co-activate canonical signalling but more effective at co-activating the Wnt/PCP pathway compared with wild type *APP*, supporting the idea of a pathologic shift from canonical to non-canonical signalling in AD that is detrimental for synapses [136].

### 4.4. Wnt Signalling, Microglia/Neuroinflammation, and AD

Microglia are the resident immune cells of the CNS [202]. Using motile processes, they constantly survey their surroundings in search of pathogens and cellular debris that need to be cleared, thus maintaining a healthy CNS environment in a tissue where extensive immune reactions, akin to what is observed in the periphery, would be detrimental. Microglia also play an important role in refining neuronal circuits both during development and in adulthood by removing or stimulating the formation of synapses as well as by removing neuronal precursors [203]. In AD however, microglia contribute profoundly to pathology. The inherent link between microglia and AD is evidenced by the fact that variants of an increasing number of genes expressed in microglia are associated with an increased or decreased risk of AD, such as variants of *TREM2* and *CD33* encoding microglial immune receptors [133,204]. The AD brain mounts an inflammatory response involving microglia (as well as astrocytes) [205]. These usually respond to CNS insults by proliferating as well as by altering their gene expression, morphology and secretomes as they transition from a senescent/surveying into an activated state, while extending their processes and migrating towards the lesion site where they initiate an innate immune response (reviewed extensively in [206]). Several RNASeq studies focusing on a range of AD mouse models have identified transcriptionally distinct subsets of microglia that emerge in and are uniquely associated with AD and have collectively been termed ‘Disease Associated Microglia’ (DAM) or ‘Activated Response Microglia’ (ARM) [122,207,208].

The transition from surveying microglia to DAM/ARM cells appears to require *TREM2* [207]. TREM2 in return signals intracellularly to affect the expression of numerous genes. In keeping, when *Trem2* was knocked out in the *PS2APP* AD mouse model, the transcript levels of 144 genes were reduced, compared to only 7 genes that were upregulated [209]. As the authors of this study stated, many of these downregulated genes were upregulated in microglia in *PS2APP* mice in the first place compared to wild type controls when both groups had intact *Trem2* [122,209,210]. This suggests that the microglial AD response involves the Trem2-regulated upregulation of numerous genes. This group of genes was enriched for positive regulators of canonical Wnt signalling according to gene ontological analysis (*Fz9*, *Sulf2*, *Bambi*, *Ptk7*, *Aspm*, and *Dkk2*) [209]. Three further Trem2-regulated genes—*Wif1*, *Ctnna3* and *Dkk2*—can be categorised as negative regulators of Wnt signalling (note that Dkk2 can be a context-dependent activator and inhibitor of canonical Wnt signalling [47]) and their biological function in this context is most likely related to negative feedback [209].

Interestingly, the existence of crosstalk between TREM2 and canonical Wnt signalling in microglia had already been discovered in an earlier mouse study [134], where it was demonstrated that the proliferative response of activated microglia not only requires TREM2 but also crosstalks with canonical Wnt signalling [134]; although not explicitly addressed within this study, it appears plausible that the same would apply to microglial proliferation in human AD brain. *Trem2^−/−^* knockout mice exhibited impaired kainic acid-induced microglial proliferation and survival, while microglial apoptosis was increased. Simultaneously, β-catenin and inactive GSK-3β protein levels were reduced. Based on these data it was proposed that TREM2 usually signals intracellularly to inactivate GSK-3β via phosphorylation by PI3K/AKT, leading to β-catenin stabilisation and transcription of pro-survival and anti-apoptotic genes. Obviously, GSK-3β and AKT are important nodal points that provide links to canonical Wnt signalling. In keeping, direct and indirect GSK-3β activation with LiCl and Wnt3a respectively rescued the microglial phenotype in *Trem2^−/−^* mice.

However, the exact function of the microglial response in the AD brain is multifaceted and whether it is regarded as beneficial or detrimental appears to be context dependent. This dichotomy can be exemplified by the way microglia respond to Aβ. It is widely established that activated microglia accumulate around Aβ plaques using the Aβ cell surface receptors SCARA1, CD36, CD14, α6β1 integrin, CD47, TLR2/4/6/9, TREM2 and that they can phagocytose Aβ (note the link between Wnt signalling and Aβ as well as TREM2 already described in earlier sections) [211,212,213,214]. This contributes to the notion that microglia play a protective/beneficial role, perhaps at certain stages of AD, even though they ultimately fail to efficiently clear Aβ in AD. Microglia form a physical barrier around the fibrillar plaque centre while limiting the halo-like leakage of the more neurotoxic oligomeric Aβ from these centres [215] (note however, that disparate findings have been published where plaque formation/maintenance was affected by microglia ablation in some cases but not in others [209,216,217,218]). In addition, microglia also orchestrate plasminogen and metalloproteinase proteolytic activities to act on Aβ [219]. These beneficial roles are complemented by the clearing of AD-related cellular debris and dead cells.

By contrast, microglia also exert detrimental effects in AD that exacerbate disease. Cytokines and chemokines secreted by activated microglia (and astrocytes) are important for the chemoattraction and response towards Aβ and cellular debris, likely helping to mitigate disease at early stages. However, chronically activated microglia continually secrete pro-inflammatory cytokines (e.g., TNFα, IL-6, IL-1α, and GM-CSF), which is an important, albeit harmful, aspect of neuroinflammation evident in the AD brain, leading to a sustained pro-inflammatory environment as well as suppression of neuronal activity and injury of bystander neurons [220]. Furthermore, microglia increasingly engage in synaptic pruning using the complement system, thus contributing to the deterioration of neuronal circuits [221,222]. All these activities are likely exacerbated due to the microglial proliferative response regulated by canonical Wnt signalling. Thus, whether microglia are friend or foe in the fight against AD may thus very well be a context-dependent matter.

The molecular mechanism underlying the microglial friend/foe dichotomy may be centred around the interaction between Wnt and Toll like receptor (TLR) signalling pathways [223,224]. However, this relationship is presently still ill-defined. Key molecular components of these pathways have been linked in a limited number of AD studies focusing on microglia, but a lot of insight into detailed cross-talk mechanisms were initially derived from peripheral immune cell or cancer responses. TLRs belong to the family of pattern recognition receptors that detect pathogenic or damage-derived molecules including Aβ, and many TLRs are expressed by immune cells such as microglia, although they are also expressed in various other CNS cells. In AD, Aβ is recognised by TLR2 and TLR4 on microglia [225,226] and in the classical TLR pathway, ligand binding culminates in the expression of pro-inflammatory genes including genes encoding IL-1β, IL-6, IL-12, TNF-α, iNOS, and COX2, driven by the transcription factor NF-κB [214]. Importantly, canonical Wnt signalling, via both GSK-3β and β-catenin, can exert both anti- and pro-inflammatory activity in non-neural models, respectively preventing and stimulating the expression of pro-inflammatory NF-κB target genes, suggesting that Wnt signalling modulates TLR signalling [227,228,229]. These divergent findings imply a context-dependent cross-talk that could explain the transition towards the pro-inflammatory state exhibited by chronically activated AD microglia. This might occur in a concerted fashion with TREM2-mediated, induced proliferation of microglia that is supported by canonical Wnt signalling [134]. In keeping, a recent study demonstrated that Wnt3a treatment ameliorated toxic microglial responses in an ischemic stroke mouse model by reducing peri-infarct microgliosis and decreasing expression of *iNos* and *TNF-α*, two known NF-κB target genes [230]. Another recent study showed in a developmental encephalopathy mouse model that the canonical Wnt agonist L803mts targeted at microglia negatively modulated microglial activation, manifesting as a reduction in *iNos* and *Cox2* mRNA levels in microglia [229]. The possibilities for Wnt/TLR cross-talk in microglia are ample and bi-directional (Wnt signalling can affect TLR signalling and vice versa) extending well beyond just GSK-3β and β-catenin although direct evidence for this cross-talk in microglia is only beginning to be discovered. The reader is referred to a previously published review for further reference [224].

### 4.5. Wnt Signalling, the Blood Brain Barrier, and AD

The blood brain barrier (BBB) is a restrictive barrier between the general circulation and the CNS serving to maintain correct brain parenchyma and cerebrospinal fluid composition by controlling influx/efflux of molecules and cells into/out of the CNS and to protect the CNS from peripheral insults [231]. In the wider context, the BBB is an essential component of the neurovascular unit that is further comprised of neurons, glia (variously including astrocytes, microglia, and oligodendrocytes), which form a tightly integrated system that ensures normal brain function. The BBB consists of endothelial cells (ECs) of the CNS microvasculature, as well as a group of perivascular components comprising astrocytes, pericytes, and the extracellular matrix. Much of the barrier function stems from two different protein structures between ECs. Tight junctions are comprised of Occludin, Claudins, and other junctional adhesion molecules assembled into a barrier forming zip-locked structure that restricts diffusion of solutes from the blood into the CNS [232]. Adherens junctions contain transmembrane cell adhesion proteins of the cadherin family (mostly VE-cadherin) that are responsible for cell-cell adhesion between CNS ECs. Intracellularly, adherens junctions are linked to the actin cytoskeleton via catenins (α-, β-, and γ-catenin), which regulate the adhesive properties of VE-cadherin (note that β-catenin can have dual roles in cell adhesion and Wnt signalling in this context), actin bundling that is linked to barrier permeability, as well as cell signalling processes [231]. To enable the passage of otherwise non-permeant solutes into and out of the CNS, the BBB is equipped with transporters such as GLUT-1 (supplying the brain with its primary energy source glucose) and P-gp (broad spectrum efflux transporter restricting solute access to the CNS or expelling solutes such as Aβ into the blood) [233,234].

Given these important roles, BBB dysfunction can have detrimental consequences on the CNS. Indeed, BBB disruptions are a key feature of AD, bringing about increased BBB permeability leading to CNS microbleeding and perivascular infiltration of neurotoxic blood-derived products, cellular infiltration and degeneration of pericytes and ECs, as well as impaired GLUT-1 and P-gp function, which negatively impact glucose homeostasis and Aβ clearance (reviewed in [235,236]). These defects may be in part linked to impaired Wnt signalling at the BBB; this is in light of the well-established role that this signalling pathway plays in CNS vasculature/BBB development and maintenance: Wnt1, Wnt3a, Wnt4, Wnt7a, Wnt7b, and Norrin have been identified as neuroepithelium derived Wnt ligands targeting the vascular endothelium that expresses cognate receptors Fz4, 6, and 8 as well as the co-receptors Lrp5/6, Gpr124, and Reck to stimulate vascular development and barrier formation specifically in the CNS via the canonical pathway [51,237,238,239,240,241,242,243]. Furthermore, canonical Wnt signalling, through transcriptional transactivation or protein-protein interactions, regulates target genes or proteins relevant for CNS vasculature/BBB function, including Claudin-3, VE-cadherin, GLUT-1, and P-gp in brain endothelial cells from zebrafish, rodents and humans [237,244,245,246,247]; this may occur in coordination with non-canonical Wnt pathways via RhoA and RhoAK-mediated inhibition of GSK-3β activation [248]. The fact that global knockout or endothelial cell-restricted conditional knockout of canonical Wnt components such as *Wnt7a*, *Fz4*, *Norrin* or *Ctnnb1* (β-catenin) in adult mice reduced claudin-1, -3, -5, and GLUT-1 protein levels accompanied in some cases by seizures, neuronal injury, haemorrhages, and inflammation demonstrates that canonical Wnt signalling is also important for CNS vasculature/BBB maintenance [249,250].

As mentioned earlier, the AD brain suffers from increased Aβ burden and impaired Wnt signalling as part of a pathological negative feedback loop (see Section 4.1). This downward spiral is further exacerbated by the fact that reduced canonical Wnt signalling strength, as brought about by Dkk1 and Aβ treatment or in transgenic mice expressing constitutively active *Gsk3b* (S9A), was shown to decrease *MDR1* (P-gp) expression in brain endothelial cells in vitro and in vivo as well as in the hippocampus of AD patients [246,251,252,253]. In absence of P-gp-mediated Aβ expulsion into the general circulation, Aβ retention within the brain is increased [254] (Note that Aβ removal from brain to general circulation involves the crossing of endothelial cells, first on the abluminal (brain side) plasma membrane, aided by LRP1, then on the luminal (blood side) plasma membrane, mediated by P-gp). This is likely exacerbated by the fact that expression of the receptor for advanced glycation end-products (*RAGE*), believed to mediate Aβ uptake from the blood into the brain, is increased [255]. Conversely, restoring *Mdr1* expression in the *hAPP* AD mouse model, significantly reduced brain Aβ levels [256].

Reduced Wnt signalling in AD may have direct consequences on energy metabolism in the brain. Canonical Wnt signalling via Wnt7a was shown to boost expression of *Slc2a1* (GLUT-1) in primary mouse brain endothelial cells [237]. With glucose being the primary energy source in the brain, reduced endothelial cell GLUT-1 levels, as observed in the AD brain [257], can lead to insufficient glucose tissue uptake. Indeed, in the *Slc2a1* deficient *APPSw/0* AD mouse model there is exacerbated AD pathology exemplified by accelerated microvascular degeneration, blood flow reductions, BBB breakdown, accelerated Aβ pathology and reduced clearance, attenuated neuronal activity, and more severe neurodegeneration [258].

Together, these lines of evidence demonstrate the importance of Wnt signalling in BBB development and maintenance. Consequently, dysregulation of Wnt pathways could contribute profoundly to the vascular and, by association, the neuro-glial, pathobiology evident in AD.

### 4.6. Are the Wnt Changes Seen in AD Simply an Exacerbated Consequence of Ageing?

The identification of genetic risk factors in the previous sections notwithstanding, age still remains the biggest risk factor for AD [112,113,259]. Indeed, a recent review has pointed out that a number of hallmarks of ageing are either also found in AD patients or can be linked to exacerbated disease in experimental models and patients [259]; these include increased DNA damage/reduced DNA repair, telomere instability, epigenetic changes, compromised mitophagy, increased senescence, stem cell depletion, increased neuroinflammation, and impaired brain metabolism. Could the same also apply to dysregulated Wnt signalling? 

We, and others in the past (e.g., [11]) have discussed that dysregulated Wnt signalling is indeed another commonality between ageing and AD, manifesting as an overall decrease in the Wnt signalling tone caused by decreased expression of Wnt ligands and receptors as well as increased expression and/or activity of negative regulators of Wnt signalling such as GSK-3β ([108]; see also Section 3.3). However, while increased expression of the endogenous canonical Wnt signalling antagonist *Dkk1* has intriguingly been identified both in old age and AD mouse models, *DKK1* upregulation has so far only been identified in AD patients, but not in normal ageing [108,119]. Comparisons between normal ageing brain and AD brain are presently difficult due to the relative scarcity of published literature on the former at the time of writing. The plethora of single cell and single nuclear sequencing studies on AD/control brain that are now emerging may shed further light on this question [260].

In spite of some apparent overall commonalities with respect to dysregulated Wnt signalling in normal ageing and AD, there is presently a gap in our understanding of the underlying molecular mechanisms and whether they are similar or conserved. It can be speculated whether the other commonalities between normal ageing and AD mentioned at the beginning of this section, might result in impaired Wnt signalling (e.g., epigenetic changes, neuroinflammation etc.). Conversely, although still far from a complete mechanistic description, links between dysregulated Wnt signalling and AD have previously been described (e.g., the connection between Aβ, CLU and upregulated *DKK1* expression [195]; see also previous segments in Section 4). It is conceivable that subthreshold pathological changes, including increased amyloidogenic APP processing, could destabilise normal Wnt signalling both in AD and normal ageing. While these changes evolve during the long prodromal AD stage, they could remain below the pathological threshold in normal ageing. This would be indicative of a quantitative difference in the relative amount of Wnt signalling dysregulation between these two scenarios. However, in absence of supporting evidence, qualitatively different molecular mechanisms are presently equally conceivable. The fact that the molecular fluid biomarker signature between normal ageing and AD is different (e.g., [184]) would support qualitative differences between the two. In this context, Wnt signalling components are yet to be firmly established as fluid biomarkers for AD, although recent studies have suggested serum DKK1 as a predictor of deteriorating disease both in AD and acute ischemic stroke patients [261,262].

## 5. Looking Forward: Does Wnt Signalling Offer Opportunities for New Therapies for AD?

Normalising dysfunctional Wnt signalling in the AD brain may represent an opportunity for novel therapeutic approaches. As has been discussed, the data points towards a decreased Wnt signalling tone and therefore a therapeutic intervention would necessitate restored signalling. This flags a potential safety concern given the extensive data on the role of dysregulated (generally an increased) Wnt signalling in cancer [263]. Similarly, Wnt signalling pathways exhibit a high degree of complexity with a plethora of cross-talk points with other signalling pathways and cellular processes, potentially making it difficult to target this signalling pathway in a controlled way without causing unwanted side effects (although some therapeutic success in experimental models has been achieved as summarised in Table 1). Thus from this perspective there is a need to proceed with caution. Generic upregulation of Wnt signalling (for instance increasing canonical Wnt signalling through use of a small molecule GSK-3β inhibitor) is unlikely to be a viable approach for treating a chronic disease. One way to proceed may be to identify ways of selectively targeting Wnt signalling (as opposed to generically increasing signalling). This could be achieved for instance by (i) targeting specific components of Wnt signalling that are dysregulated in the disease, or (ii) by targeting signalling pathway components that are selectively expressed in specific cell types. Additionally, while as has been discussed (Section 4.1) there is an evidence base to support the modulation of APP processing/β-amyloid production by Wnt signalling, from a therapeutic perspective it can be argued that there are more direct ways to target the β-amyloid cascade [264], and the lack of success in clinical trials to date, has led to a general deprioritisation of β-amyloid as a target [265]. As such, this will not be considered further.

The secreted Wnt antagonist DKK1 is upregulated in AD and directly implicated in synapse dysfunction and loss (see Section 4.3); as such it is a potential therapeutic target. One approach that has been investigated in the context of another chronic disease, osteoporosis, is the use of an anti-DKK1 monoclonal antibody to neutralise its inhibitory activity; indeed, this has been shown to be efficacious in animal models [266]. The obvious caveat for the application of this approach to AD is our current inability to deliver large therapeutic molecules such as antibodies across the BBB. A second approach would be to develop a small molecule drug that could inhibit the binding of DKK1 to its cell surface co-receptor LRP6. However, developing small molecule drugs that inhibit the interaction of large proteins has generally proven to be very challenging [267]. The third approach may be to employ antisense oligonucleotide (ASO) technology to target DKK1. ASOs are in development for a number of disorders including neurodegenerative diseases such as Huntington’s disease [268].

In addition to considering the therapeutic modality, there are some other points to consider. While *DKK1* expression is increased in AD brain (Section 4.3), the stage of the disease when this occurs has not been defined. Such understanding is essential to inform when to intervene with a DKK1 based therapeutic. Similarly, can DKK1 be detected in cerebrospinal fluid (CSF)? Further establishing DKK1 CSF or serum biomarkers [261,262] would be valuable in understanding where DKK1 fits in the pathophysiology of AD, as well as being a target biomarker for evaluating DKK1 therapeutics. It would also have the potential to enable precision medicine by identifying those patients who have upregulated *DKK1*.

Microglia and neuroinflammation have become an area of significant activity in the last few years, including as potential targets for therapeutic intervention in AD. A particular focus has been the TREM2 signalling pathway, where there are efforts being explored to stimulate this pathway and as a therapeutic approach for AD [269]. We have discussed above (Section 4.4) that there are a number of studies describing an interaction between TREM2 and Wnt signalling in microglia. While this offers the potential for modulating TREM2 signalling via Wnt, it is premature from a therapeutic perspective. Our understanding of the interaction between these signalling pathways is currently limited. Importantly, there is currently no data as to whether this occurs in human microglia, and whether there is dysfunction or dysregulation in AD that could contribute to the pathology of the disease.

In Section 4.5 the importance of the Wnt signalling system in maintaining a functional BBB was discussed, as was the dysfunction of the BBB in AD. While the loss of Wnt signalling in brain endothelial cells contributes to BBB breakdown in disorders such as multiple sclerosis [270], hemorrhagic stroke [271], and traumatic brain injury [272], no such evidence is currently available for AD, either in animal models or human disease. This is clearly a gap in our knowledge, and a gap in terms of supporting the notion of modulating BBB Wnt signalling as a therapeutic approach for AD. Nevertheless, given the decreased expression in AD brain of Wnt regulated genes essential for normal BBB function such as endothelial cell tight junction proteins and transporters, it is a plausible hypothesis that increasing Wnt signalling tone at brain endothelial cells has the potential to restore expression and therefore be of therapeutic benefit. Indeed, the Wnt co-receptor Gpr124, through which endothelial cell signalling is mediated [241,242,243] is selectively expressed in brain endothelial cells and so in principle could be targeted to generate agonists or positive allosteric modulators, as has been done for other members of the GPCR gene super-family [273].

In summary, while our understanding of the dysregulation of Wnt signalling in AD has increased, there remain significant gaps, particularly a clear understanding of the dysfunction in the human disease, and how this links to the pathophysiology. This deeper knowledge will be essential in informing rational approaches to developing new therapies that could become part of our armoury for treating AD in the future. However, the success of future Wnt-therapies will inevitably require early intervention and hence early detection at pre-clinical stages. Decreased Wnt signalling strength is evident in the ageing brain and likely further declines in the transition to pathological ageing, mild cognitive impairment, and AD. Future research efforts should thus also focus on the development of early detection methods for reduced Wnt signalling strength, as proposed in this section for DKK1, in potential future LOAD patients at prodromal stages [274,275].

## Figures and Tables

**Figure 1 brainsci-10-00902-f001:**
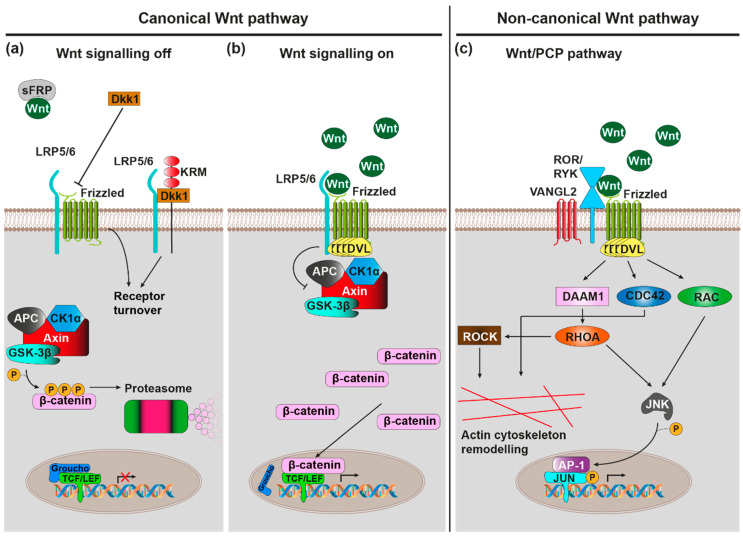
Wnt signalling pathways. Note that emphasis is placed on pathways and components that are of greater relevance in the context of AD. (**a**) Canonical Wnt signalling pathway, off: In absence of Wnt ligand binding to Frizzled (Fz)-LRP5/6 heterodimeric cell surface receptors, the Axin/APC/CK1α/GSK-3β destruction complex phosphorylates the transcription factor β-catenin, marking it for proteasomal degradation. Endogenous canonical Wnt antagonists (including sFRP and Dkk1) further favour the ‘off’ state. (**b**) Canonical Wnt signalling pathway, on: Wnt ligand binding to Fz-LRP5/6 recruits the scaffolding protein DVL, which in turn inhibits GSK-3β activity of the destruction complex. β-catenin subsequently accumulates in cytoplasm, allowing it to translocate to the nucleus where it transactivates the TCF/LEF-mediated expression of canonical Wnt target genes. (**c**) Wnt/planar cell polarity (PCP) pathway: Wnt ligand binding to Fz-ROR/RYK co-receptors recruits DVL, which in turn activates the small GTPases RHOA, RAC, and CDC42. RHOA and RAC conjointly activate JNK, which stimulates Jun transcription factor dependent gene expression via phosphorylation. Furthermore, RHOA, via ROCK, and CDC42 stimulate actin cytoskeleton remodelling.

**Figure 2 brainsci-10-00902-f002:**
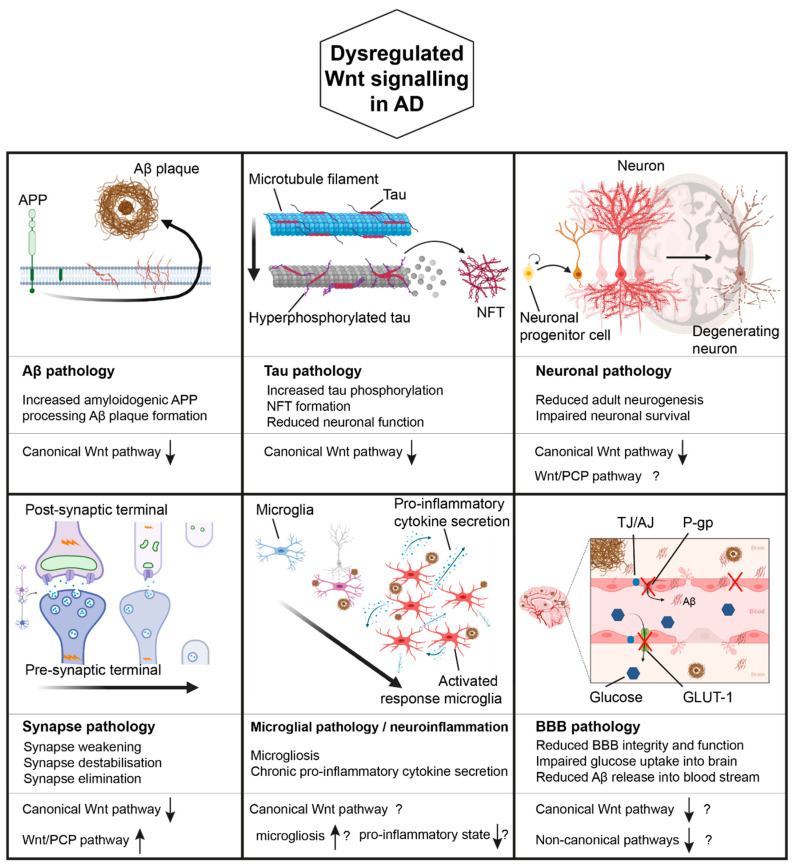
Wnt signalling in AD. Dysregulated Wnt signalling may contribute to various pathological manifestation of AD, including Aβ pathology, Tau pathology, neuronal pathology, synapse pathology, microglial pathology/neuroinflammation, as well as BBB pathology. Known or hypothesised (indicated by ‘?’) changes in Wnt signalling associated with each manifestation are indicated (based on published literature). ↑ and ↓ indicate increased and decreased Wnt signalling respectively. Created with Biorender.com.

**Table 1 brainsci-10-00902-t001:** Summary of Wnt targeting approaches in the published literature, where an increase of or restoration to normal signalling levels was attempted to treat neurological disorders (predominantly AD) in experimental models. ‘↑’ and ‘↓’ indicate increases or decreases in a measured experimental outcome.

Reference	Type of Manipulation	Experimental Model	Experimental Outcome
Parr et al., 2014 [153]	Wnt3a, β-catenin over-expression	N2Asw murine neuroblastoma cell line	↓ Aβ levels, ↓BACE1 levels
Alvarez et al., 2004 [157]	Wnt3a treatment	Rat primary hippocampal neurons (treated with Aβ)	↑ neuronal survival, ↓ GSK-3β phosphorylation, ↓ tau phosphorylation, restored cytosolic β-catenin levels, ↑ increased expression of Wnt target gene engrailed-1
Zhang et al., 2019 [230]	Wnt3a	Ischemic stroke (tMCAO) mouse model (in vivo)	↓ infarct volume, ↓ neurologic deficits, ↓ peri-infarct apoptosis, ↓ microglial activation, ↓ pro-inflammatory cytokine secretion (iNos, TNFα), ↓ astrogliosis
Zheng et al., 2017 [134]	Wnt3a, LiCl, TDZD-8	Trem2-/- mouse primary microglia	↑ β-catenin levels, ↑ microglial survival, ↑ microglial proliferation
Cerpa et al., 2010 [190]	Wnt5a	Rat primary hippocampal neurons and hippocampal slices (treated with Aβ)	↑ fEPSP & (AMPAR, NMDAR) EPSC amplitudes, ↑ post-synaptic PSD-95 clusters
Marzo et al., 2016 [111]	6-BIO (GSK-3β inhibitor)	Rat primary hippocampal neurons (treated with Dkk1)	↑ excitatory synapse numbers
Salcedo-Tello et al., 2014 [175]	6-BIO	Rat hippocampal slices at 3 months (young) and 18–20 months (aged)	↓ GSK-3β activity, ↑ β-catenin levels
Quintanilla et al., 2005 [158]	17β-estradiol, Trolox (antioxidant)	Rat primary hippocampal neurons (treated with Aβ)	↑ neuronal survival, ↓ endoperoxide production, ↓ GSK-3β activity, ↑ cytoplasmic β-catenin, ↑ Wnt5a & Wnt7a levels (17β-estradiol), ↓ Ca^2+^ mediated toxicity
	Wnt7a		protection against Ca^2+^ mediated toxicity
Magdesian et al., 2008 [159]	cSP5 (Aβ-binding peptide)	N2Asmurine neuroblastoma cell line (differentiated)	↑ total & nuclear β-catenin
Elliott et al., 2018 [136]	Fasudil (Rock inhibitor)	Rat primary cortical neurons (treated with Dkk1)	Protection against Dkk1-mediated dendritic spine loss, and Aβ production
		3xTg AD mice (in vivo)	↓ soluble and insoluble Aβ
Vargas et al., 2014 [187]	FOXY-5 (peptide)	APP/PS1 mouse model (in vivo & hippocampal slices)	Rescued memory retention, ↑ fEPSP, rescued LTP deficit
Van Steenwinckel et al., 2019 [229]	L803mts (Wnt/β-catenin activator; targeted to microglia by 3DNA)	Mouse developmental encephalopathy model (IL-1β-mediated; in vivo)	↓ microglial expression of pro-inflammatory cytokines (iNos, Cox2), restored myelination, restored short- & long-term memory retention
De Ferrari et al., 2003 [156]	LiCl (GSK-3β inhibitor), Wnt3a, PMA (PKC activator)	Rat primary hippocampal neurons (treated with Aβ)	↑ neuronal survival, restored cytosolic β-catenin levels
	LiCl	Intrahippocampal injection of Aβ fibrils in rats (in vivo)	↑ neuronal survival, improved deficit in spatial learning (Morris water maze) induced by Aβ
Scali et al., 2006 [176]	LiCl	Rats (3 months) injected with recombinant human DKK1 (in vivo)	↓ DKK1-stimulated GSK-3β activity, protects against DKK1-mediated neurodegeneration, tau hyperphosphorylation, and astrogliosis
Ross et al., 2018 [200]	miRNA-431 (silences Dkk1 receptor Krm1)	Mouse primary cortico-hippocampal neurons from 3xTg AD mouse model (in vivo)	↓ Aβ- and Dkk1-mediated synapse loss, ↓ neurite degeneration
Vargas et al., 2015 [154]	WASP-1 small molecule	Rat primary hippocampal neurons	↑ synaptic transmission, rescued hippocampal LTP impairments due to Aβ
		APP/PS1 mouse model (in vivo)	↓ synaptic protein loss, ↓ tau phosphorylation
	Y27632 (Rock inhibitor)	Rat primary hippocampal neurons (treated with Dkk1)	↑ excitatory synapse numbers
Sellers el al., 2018 [201]	Y27632, Fasudil	Rat primary cortical neurons (treated with Aβ or Dkk1)	Protection against Aβ- or Dkk1- mediated spine density reduction, ↑ PSD-95 puncta, ↑ GluA1 puncta
	Fasudil	Rats (treated with Aβ; in vivo)	Protection against Aβ-driven cognitive impairment, ↑ performance in NOR task
	LiCl	Trem2-/- mice (in vivo)	↑ microglia cell # in hippocampus and cortex, ↑ β-catenin, ↓ GSK-3β activation, ↑ cell cycle markers (c-Myc, Cyclin D1)

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
