# Peer review of "Dysregulated Wnt Signalling in the Alzheimer’s Brain"

_brainsci, 2020, doi:10.3390/brainsci10120902_

Round 1

Reviewer 1 Report

This is a very comprehensive article.  As the authors appear to have proposed that Wnt signaling is at the hub of all AD pathological processes (not sure l would agree about the way this is described for neuroinflammation), it is clear they believe pharmacological manipulation of wnt signaling is a viable approach.  So could they maybe include a section summarizing the studies of consequences of chemical or genetic manipulation of wnt signaling in different neural cell types or experimental animals, and could they discuss which are the promising candidates (if any).  Some of this information can be gleaned from other sections but putting it into a concise section would be interesting.

Minor point.  In the studies being reviewed, could they make it clear which cell types are the focus of the study.  This is only clear in a few examples.

Reviewer 2 Report

The topic of this review is timely and essential because Wnt signaling deficits in AD are becoming increasingly relevant, yet receive little attention in the field. While the authors attempt to address this important and complex topic, the review falls short of expectations because it contains too many incidental details that detract from its central theme. As a result, this review reads like a  report or isolated overview of information, rather than an academic review paper that synthesizes the research and unites important concepts into a theme of why Wnt signaling deficits should be an important focus for understanding and treating AD.  I feel that with significant revision and some re-organization, the authors can achieve this and encourage them to do so while addressing the following specific concerns:

1. The use of the word “deregulated” in the article title is not the proper term; “dysregulation” is. Deregulated means “unregulated” which is not the case of the Wnt pathway in Alzheimer’s disease. In fact, canonical Wnt signaling in AD is suppressed which is quite the opposite of “deregulation.”  The correct term to use is dysregulated, which means “dysfunctional or aberrant regulation” and more clearly describes what is happening to this signaling pathway in AD.  

2. The first paragraph should grab the reader's attention about the significance of the paper, and why this review (among all the other literature reviews on this topic) is significant and adds new information/interpretation to the current state of the research in the field. The writing in this paragraph sounds "choppy" and fails to grab attention. In line 25, replace the term “carers” with "caregivers", as the former is not as common of a term as the latter. Thus, I would recommend rewording that entire sentence to read ... “profoundly impacts caregivers, families, and society as a whole” ...to better represent AD’s significance. Also, the last sentence (lines 30-32) seems to have some syntax errors and should be rewritten to flow more smoothly.

3. There are numerous published manuscripts with elegant visual depictions of Wnt signaling cascades and it might be best to replace Figure 1 with a reference for one of these previous works.  Figure 1, as presented, is cumbersome with too many detailed labels using small font and doesn’t clarify the pathway any better than the written description. The goal of a figure is to summarize and enhance what is written in the text—this figure does not accomplish this. Given that so many other manuscripts do provide graphic representations of normal Wnt signaling pathways, the authors would be better off creating a figure that highlights the specific elements of Wnt pathways that are shown to be altered in AD, as this would support the review’s purpose and be informative to the reader. Therefore, the recommendation is to either remove this figure or replace it with something of this nature. 

4. The authors spend several paragraphs describing the detailed role of Wnt signaling role in the developing brain, but this doesn’t seem to provide any useful information for interpreting any of the later discussion about Wnt defects in AD. This creates a problem because all of this superfluous detail distracts from the purpose of the paper which is to link Wnt deficits with AD. While Wnt patterning is involved in hippocampal development and synaptogenesis, these facts can just be stated succinctly because the authors do not provide any details of how these organization processes are pathologically changed in AD (because they aren’t necessarily). Therefore, I recommend reducing the section on Wnt signaling in developing brain into perhaps a few summarized sentences acknowledging Wnt’s role in neural development.  As for the sections on Wnt’s role in the mature and aging brain, this information seems like it would be best re-organized and presented in context with how Wnt signaling differs or is altered in comparison to the AD brain. Are the age-related decreases in canonical Wnt signaling qualitatively different than what is seen in AD, or just quantitatively different (i.e. an exacerbating of the normal aging process). The distinction between changes associated with normal, healthy aging and the pathological disease state of AD should be emphasized.  

5. The statement written in lines 310-311 about “boosting” Wnt signaling should be rewritten because it comes across as oversimplified: does this mean “restoring” deficient signaling or “enhancing” basal signaling? The act of enhancing Wnt signaling is no small endeavor given that most therapeutics are designed to suppress it (i.e. anti-cancer drugs), and the statement oversimplifies the complexity of determining which of the many mechanisms of the Wnt cascade are most strategic to manipulate in terms of enhancing/restoring Wnt signaling.

6. While Figure 2 has the title “Deregulated wnt signaling in AD,”  there is no description of which specific Wnt signaling elements are associated with each of these pathological problems of AD and therefore, the figure is uninformative, and should be revised accordingly. Also change the term “deregulated” to “dysregulated” in any revisions of the figure. 

7. Its unclear what message the authors are trying to convey with what is written in lines 460-464 about frontotemporal dementia. This distinction between FTD and AD here doesn’t seem relevant to the introduction of the topic of tauopathy. Also, the statement made in line 465 is outdated and not supported by current research in the field, or even by the more recent papers cited by the authors. The overall consensus of research in the field does not consider tau pathology to be necessary for amyloid-beta pathogenesis in either human clinical cases or rodent models. In fact, there is more evidence that amyloid beta facilitates the development of tauopathy, although this topic itself is contentious and amyloid-independent mechanisms for tauopathy pathogenesis have also been identified. Therefore, the authors should reword this statement/section to reflect the fact that there are important mechanistic linkages between amyloid beta and tau but the precise nature of the  relationship is complex.

8. In lines 497-498, the authors state that amyloid beta formation precedes synaptic dysfunction, cognitive deficits and tauopathy, and this is also a  contentious and outdated finding (the Sperling report is from 2011). Some of the earliest pathological indicators in human tissue are actually in fact tauopathy, again, underscoring the independent pathways by which each of these pathologies arise, and some of the most recent clinical findings show that plasma or CSF levels of tau180 are the earliest biomarkers (Rodriguez + 2020: PMID: 32720099). There is nothing wrong with linking amyloid aggregation to synaptic dysfunction, but one must be careful in describing the nuances, direction, and timing of the mechanistic association between these events as it is not well-established in the field. 

9. There are numerous reviews on Wnt signaling deficits in AD, and the authors should strive towards the goal of distinguishing this review as unique and useful to someone who investigates this topic (such as myself). As such, the discussion/conclusions section of a review should be a call for action based on  the data that has been reviewed and what is clearly still needed to move this field forward. This section seems to fall short of this. The authors mention  concepts such as "early detection" of Wnt deficts in AD but fail to expound upon why/how this can be accomplished and what work still needs to be done. While other sections of this manuscript provide too many details, this section does not provide enough detail to drive home the importance of what this review of information on Wnt signaling in AD tells us about the state of the field. 

Round 2

Reviewer 2 Report

The authors have done a commendable job revising this manuscript and responding to reviewer feedback. The manuscript is much improved and provides a thorough, solid, and interesting review of this important topic. Great job!